# A Transition towards a Food and Agricultural System That Includes Both Food Security and Planetary Health

**DOI:** 10.3390/foods12010012

**Published:** 2022-12-20

**Authors:** Maria Hofman-Bergholm

**Affiliations:** 1Research and Development, Entrepreneurship and Wellbeing, Centria University of Applied Science, 67100 Kokkola, Finland; maria.hofman-bergholm@centria.fi or mhofman@abo.fi; 2Faculty of Education and Welfare Studies, Åbo Akademi University, 20500 Turku, Finland

**Keywords:** food systems, biodiversity, sustainable development goals, food security, healthy diets, planetary health, agricultural praxis, farming, mindset

## Abstract

This theoretical paper builds on a multidisciplinary framework which is structured to acknowledge the need to combine different research disciplines to understand the problems within our current unsustainable food system and be able to develop possible solutions through new innovations. Current food production methods come at an environmental cost as they generate large amounts of greenhouse gas emissions which affect biodiversity and climate change. The article shows that the problems surrounding food systems and our culture around food, are multifaceted and intricate. The fact is that a growing number of citizens suffer from obesity with various consequential diseases as a result, while a part of the population is still malnourished and dying of hunger. This paper summarizes results from some fairly new studies and different international policy reports to try to clarify how broad the problem is, which is crucial to find new pathways forward to address the problems. Through theoretical discussion, the paper identifies some of the deep underlying root causes and fundamental reasons as to why the urgent needed change is so slow.

## 1. Introduction

In September 2015, the world’s leaders adopted a new development agenda and global goals for sustainable development (SDG). Agenda 2030 consists of 17 global goals and 169 targets for sustainable development that aim to i.a., eradicate poverty, stop climate change, end hunger, create peaceful and secure societies and halt the biodiversity loss [1]. Agenda 2030 is the most ambitious plan for creating sustainable development that the world has ever adopted, and now at the end of 2022 we have reached about halfway to 2030 from the publication year, 2015. 

In 2020 the United Nations (UN) held an SDG Moment Meeting for world leaders and asked Ola Rosling to summarize the progress across the SDGs and their targets since 2015. Rosling, presented information about 35 of the targets as the other targets did not have sufficient updated numbers to track global progress. The presentation revealed that only six of the targets were on track in 2020, the progress of 24 targets was indicated to be too slow or standing still and five of the targets had gone in the wrong direction [2]. The targets going in the wrong direction were climate change, biodiversity, food security, slum dwellings and unsentenced prisoners. Regarding climate change, it must be called a major defeat as the UN has worked on climate change since 1992 when the first convention was adopted, “United Nations framework convention on climate change” [3] and still it is not under control. 

The Sustainable development goals report 2022 [4] reveals that about 828 million people suffered from hunger and about 2.4 billion people lacked regular access to adequate food in 2021. Food insecurity and hunger are obviously also still significant problems in the world. Statistics show that it is also a fairly unequal world as a significant proportion of the population is starving at the same time as about 30% of all food produced is never eaten [5,6], and more than 39% of adults are overweight or obese in 2016 [7]. Food waste and global food production counts for up to 37% of the global greenhouse gas emissions [8,9,10,11], and it is a fact that our current food system is the primary driver of the world’s biodiversity loss [5,6,12]. This clearly shows that we have a profound problem in our current global food system. To produce food with current methods, land is degraded, and huge amounts of water, energy, fertilizer and pesticides are used [5,6]. This all comes at an environmental cost [13] and a large share of the global GHG emissions, which are connected to climate change [14]. Put this way, the first mentioned target that has gone in the wrong direction according to Rosling’s presentation [2] are all linked closely together; food system, biodiversity loss and climate change. It emerges that this is an important area where new innovations and more research should have the highest priority as a necessity to slow down climate change, and yet provide food and health for the whole world’s population. 

The objective of this study is to recognize and highlight known problems within our current global food systems, identified in recently published research and reports. The article also discusses different suggestions on how to address some of the known problems found in the literature. This approach has been chosen, as recent research points out multidisciplinary issues in the current food systems, but still the development seems to be heading in the wrong direction. The hypothesis this theoretical article builds on is that a multidisciplinary approach, collaboration, and knowledge exchange across the boundaries between different subjects and professional orientations could help in bridging the knowledge gap that seems to cause the continued “business as usual” attitude in society.

In this theoretical article the relation between climate change, biodiversity and food security will be illuminated and discussed. This paper will take a multidisciplinary approach focusing on food and health issues. The article will summarize reports and research results from different disciplines as the problem with our food system appears to be multifaceted, and there is a need to include multiple disciplines to find pathways forward. The last subsection will provide some ideas around innovative projects in Finland. 

## 2. Method

This study is based on a non-structured qualitative analysis performed through a traditional literature review. The flexibility in traditional literature reviews allows the method to be useful in synthesizing knowledge beyond research findings, such as theories, practices, and policies. Using qualitative research of literature as a method, there are no requirements on quantity or the completeness of the material. Instead, the interests of the research are in focus. Differences and similarities between things are compared through the researcher’s reasoning, research is summed up, and it grasps the rules of things [15].

## 3. Why Biodiversity Counts and How It Sticks Together with Climate Change and the Food System

Ecosystems and biodiversity are woven into many of the SDGs and their associated targets as they have a direct contribution to human well-being and development [16]. The health of the world’s population is dependent on natural systems that provide the population with ecosystem services such as water, food, energy, carbon sequestration and building material [5,14]. Biodiversity is also regarded as the center in many economic primary sector activities, such as agriculture, forestry and fisheries and in this way sustainable development, biodiversity and ecosystems are interconnected to macro economy [16,17]. This is how the European Environment Agency defines the word biodiversity:

“*Biodiversity is the name given to the variety of ecosystems (natural capital), species and genes in the world or in a particular habitat. It is essential to human wellbeing, as it delivers services that sustain our economies and societies.*”[18]

It is obvious that the economic values of European culture dominate in the Western concept of sustainability [17]. Currently, the world´s biodiversity is declining at an alarming rate everywhere, which is a serious situation as the wellbeing of humankind is dependent on nature [5,6,14,19,20]. Here, it would be important to raise an understanding that it is not only a question about economic welfare, far more important is to understand that the biodiversity is essential to food and agriculture [14,19], if there is no biodiversity there will be no adequate food to eat [21]. 

Nowadays, it is quite common knowledge that climate change affects and increases the loss of biodiversity [19] because species do not have time to adapt to the warmer environment [22]. At the same time, there are undesired species such as invasive plants and animals and crop pathogens thriving in an unbalanced ecosystem. Lesser-known facts, but all the more important, are that today’s global food system is one of the major reasons we are destroying biodiversity [14,19,20]. The development over the past 50 years, of an accelerating conversion of natural ecosystems for crop production or pasture is found to be the biggest driver of habitat loss [6]. This agricultural practice is driven in an unsustainable manner. The ongoing conversion of natural ecosystems for crop production or pasture decreases biodiversity, pollutes water and air and depletes freshwater [5,6]. It is not only the changes in the use of land that reduces biodiversity, but conventional farming methods also use a large amount of fertilizer and pesticide which pollutes water, soil and air [5,6,14].

Recent studies imply that the intensity of agriculture creates homogenous landscapes harmful to biodiversity, and for example, there are studies indicating that the use of neonicotinoid pesticides in agriculture may have a connection to the recent decline of honeybees [23]. Pollinators and other invertebrates and micro-organisms e.g., insects, birds, earthworms, soil-dwelling fungi, and bacteria, which are in decline, are all vital to humans as they keep soils fertile, pollinate plants, purify water and air, fight crop and livestock pests and diseases. Without these assistants from nature, agriculture would be impossible [23,24].

A recent report “One Health Joint Plan of Action (2022–2026). Working together for the health of humans, animals, plants and the environment” [25] argues that among other drivers we have an unsustainable agricultural production and an ongoing intensification of it, we have largescale deforestation and land degradation that are all threatening the ecosystem’s functions and constitute an increased health risk, especially among the most vulnerable communities. The relationship between natural systems, food production, and health are affected by the effects of environmental degradation and the erosion of ecosystem services. That is why a transformation of the interactions between us humans, the plants, the animals and the environment we all share is urgently needed. The invisible bond between us must be reassessed and balanced to ensure human, animal and plant well-being and health, and it is crucial for the SDGs to be achieved [25]. 

Biodiversity is essential to food and agriculture and indispensable to food security, sustainable development, and the supply of many vital ecosystem services. There is a call for more multidisciplinary research on food and agricultural systems, with more focus on interactions between different components of biodiversity for food and agriculture [14]. This reveals how important biodiversity is for our survival and why the decline of biodiversity needs to be halted. As our global food system is designated a driver of biodiversity loss, there is a need to take a closer look at it. 

## 4. Our Current Food System and Its Impact on Nature 


*The EAT-Lancet Commission on Food, Planet, Health brought together 37 world-leading scientists from across the globe to answer this question: Can we feed a future population of 10 billion people a healthy diet within planetary boundaries? The answer is yes, but it will be impossible without transforming eating habits, improving food production, and reducing food waste.*
[26]

It is an undauntable fact that today’s global food system still fails in providing nutritious food to a big part of the world’s population. According to The Sustainable Development Goals Report 2022 [4], it was estimated that in 2021 about 828 million people suffered from hunger. In 2021 about 2.4 billion people lacked regular access to adequate food (Ibid.). With this in mind, it is terrible to realize that the current global food system has shown to be inefficient, as about 30% of all food produced is never eaten [5,6,11] and in addition, there is research from the IPCC [9] showing that food loss and waste is responsible for about 8–10% of the total global greenhouse emissions [11,27]. 

For people to understand the size of this emitter it needs to be put in perspective with for example aviation, which a lot of people are aware of being a GHG emitter, but aviation only counts for about 2.5% of the total GHG emissions according to “Our World in Data” [28]. This means that aviation is a relatively low emission source in percentage terms compared with food waste, but then you have to consider the fact that, according to Westlake [29] among others, 95% of the global population has never been on a plane. Seen from this point of view, a small part of the world population accounts for a fairly large part of emissions through flying. Still less discussed in the public debate, are the emissions from our food production and food waste. In “The Living Planet Index 2022” [20] there is a figure showing humanity’s ecological footprint from activities and the biggest print (30%) comes from “food” activity. According to an UNEP report [11] our global food system counts for as much as 37% of the world’s greenhouse gas emissions, from what we eat, what and how we grow, the forest we clear, and the food we throw away [8,9,11].

As the ongoing deforestation is due to the process applied to implement pastures or to certain parts due to crop production as feed for cattle, meat consumption is one of the implied drivers of biodiversity loss and greenhouse gas emissions. According to “The State of the World’s Biodiversity for Food and Agriculture” [14], livestock production chains are estimated to be responsible for about 14.5 percent of anthropogenic greenhouse-gas emissions. Joseph Poor and Thomas Nemecek [30] performed a large meta-analysis study on the climate footprint of food in 2018. In their study Poor and Nemecek [30] found that of all the goods analyzed, beef coming from cattle has the absolute highest climate footprint. Their meta-study revealed that the production of one kilogram of beef leads to 99.48 kg of CO_2_e (Carbon dioxide equivalents, abbreviated “CO_2_e”, is a measure of greenhouse gas emissions). 

It is important to point out that this does not apply to all countries’ meat production as the praxis of production and production systems differs between countries. According to FAO [14] the livestock production systems can be grassland-based or landless based. Virkajärvi and Järvenranta [31] state that in terms of greenhouse gas emissions, they estimate that Finland’s importance and share are somewhat smaller than the global harm caused by cattle farming. The emissions per individual cattle beast are as harmful as elsewhere in the world but there are several factors recognized that mitigate the overall harm. The footprint of meat production differs regarding, for example, what the cattle are fed with and product efficiency, in other words if there are also other products coming from the cow. There is also a difference if the cattle walk outside or live their whole lives indoors. Compared with Poor and Nemechek’s [30] meta-analysis that found one kilogram of beef counts for 99.48 kg of CO_2_e, the Finnish Natural Resources Institute LUKE calculated that one kilogram of domestic beef leads on average to approximately 17 kg of CO_2_e [31]. This is due to factors mitigating the harm. Finnish cows are used product efficient; they are used both as meat and milk producers [32], and they are not fed on imported soy [31,33], which is a major global environmental and climate problem as huge amounts of forest are cleared to make way for soy farms for animal feed [34]. 

In many countries, cattle are mainly fed grains suitable for human consumption such as soy and maize [14,34]. This leads to a tension between different purposes for the land use, is it used to grow crops for feeding humans, or crops for feeding livestock [34]? If the cattle instead eat grass (meadow) grown on their own farm, and some grains, crop and silage produced on the farms or purchased direct from other nearby farmers a circular feed system could replace cereals and soy (ibid.). For example, the Finnish cattle mainly eat grass and some grains and rapeseed [33,35,36], and that is one of the reasons why the climate impact is lower from Finnish cattle than in some other countries. Grass eating, outside walking cows both lower the carbon footprint and increase biodiversity. Long-term grass and hay cultivation imply that the land is covered with plant cover even in winter, which is good for the climate because it binds carbon to the soil [37,38]. Meadows are also important for biodiversity. A lot of insects and plants depend on the presence of meadows, grazing animals and the animal’s feces. Many invertebrates are associated with herbivore dung, which means that many species historically found in meadows with grazing cattle in cultural landscapes, have diminished or are close to being extinct [39]. 

This reasoning indicates that production methods and praxis in agriculture are of great importance for the meat’s environmental and climate impact. Among others Haines and Frumkin [5], and the EAT-Lancet Commission summary report [12] argues that there are three different categories of solutions for the problems with our current, unsustainable, food system. The researchers focus on solutions that at the same time are fruitful for human health and for the health of the planet. The three suggested categories are (1) dietary changes, a global shift toward healthy diets (2) changes in agricultural technologies and practices, improved food production and (3) reduction in food loss and food waste. 

Biodiversity underpins our economy, society and ultimately our survival as a species in the long run. If we don’t manage to stop the loss of biodiversity it will undermine the basis of our access to food, health, and quality of life [22,40]. 

## 5. Dietary Changes and a Change in Agricultural Practices Could Pave the Way for Biodiversity and Health

The planet has been altered more by agriculture than any other human activity and now we are experiencing a dangerous decline in nature which humans are causing. There is an urgent need to transform our current food systems to become more sustainable and resilient, in order to reverse the ongoing environmental degradation, to restore ecosystems and to ensure food and nutritional security [41]. For a redesigned and more sustainable food system, there is a need to change the patterns of demand for food. In 2020 a major study [42], concluded that roughly one portion of red meat a week, with the rest of the meals being totally vegetarian, or with some fish dish from local stocks if we want more meat, is what we should aim for. 

A number of policy documents and reports argue for a shift towards more plant-based diets in rich countries [5,6,8,9,10,12,43], but to change existing habits is not easy as there has been many joint efforts over the last decades to create an always growing demand for different goods [44], most likely also including food. Wolff et al. [44] argues that “talented consumerism architects have succeeded in shaping norms, values and narratives that attract buyers to choose a lifestyle where they express themselves through consumption.” [44] (p. 11). 

To re-shape these values and norms towards a planetary health awareness is possible but not easy. It will need to include the work of media experts, marketers, business leaders, policymakers, and other stakeholders. The United States can state an example as during the past years there has occurred a fall in consumption of red meat due to a growing awareness of health consequences and environmental consequences of meat consumption [23]. This has encouraged the food industry to invest in research and development to be able to respond to this growing demand of alternative products to meat. This has resulted in new carefully branded, alternative products with competitive prices and it appears that population-wide shifts can be supported by a combination of altered behavior and a change in the retail priorities (Ibid.). 

Just like the official “meat debate”, which has reached large numbers of people through the media, it would be important to raise ordinary people’s knowledge about the different ecological footprints of different consumer goods, knowledge about the fact that it is not only meat that has a negative impact on the climate. The alternative products produced in the wake of a reduced meat consumption do not automatically need to be healthy to humans as the products might be ultra-processed [23], which according to da Silva et. al. [45] also have a very high climate footprint. Plant and vegetable production do not automatically have to be sustainable either. Cultivation of plant-based products, fruits, and other crops can be problematic globally, given e.g., water use, erosion, clearing of rainforest, negative impact on biodiversity or social aspects [6].

To address issues like this, there are some researchers that suggest a shift towards an agro-ecology with regenerative or organic farming. If we manage a reduction in food waste and a shift towards a more plant-based diet (it would lead to a reduction in farmed animals and feed crop production), organic agriculture could feed more than 9 billion people in 2050 [6]. Unfortunately, organic farming is known to produce lower yields [5,6,13,46] which is a problem as more land needs to be used to produce the same amount of food as intensive conventional farming. Haines and Frumkin [5] refer to a couple of surveys and a meta-analysis trying to find evidence for the benefits of organic farming. They found that organic agriculture in industrialized countries using current practices does not contribute to a food system with a reduced environmental footprint, mainly due to the increasing land use it demands. But they also point out that there might be other benefits with organic farming that are less easy to measure like greater biodiversity and lack of pesticide residues in the environment and in the food produced. (Ibid.)

However, many researchers seem to agree that an increase in crop yield is necessary but also dietary changes. According to eg., Pörtner et al. [10] an intensification in agriculture must be done sustainably or the detrimental effects on the environment of the intensification will outweigh the benefits. One important thing Pörtner et al. [47] raise for improving soil biodiversity and health is reducing tillage, reducing the use of pesticides and increasing organic material input. They also argue for a reduction in the use of fertilization in agriculture as excessive fertilization results in N_2_O emissions which are a potent GHG. Verdi et al. [48] found out that the most promising strategies to maintaining a high agricultural production is managing to increase the yield in organic farming and reduce nutrient losses/emissions from conventional farming. As there is, according to FAO [14] (p. 3) “abundant evidence that intensification of crop, livestock and aquaculture systems through excessive use of synthetic inputs adversely affects BFA and particularly associated biodiversity”, we must re-think the thought of more intensification in our current praxis of farming. A regenerative agricultural approach is not focusing on doing less harm but on creating healthy, resilient systems. A regenerative agricultural approach works in alignment with living systems, and it could revitalize the soil, the water, the flora, the fauna, livelihoods, cultures, and planetary health [8]. 

Pörtner et al. [10] argued that a change towards plant-based diets and a reduction in food loss and food waste would affect both climate change and biodiversity loss in a positive direction. Here for example Haines and Frumkins [5], suggestions concerning changes in agricultural technologies and practices could be fruitful in addition to dietary changes. According to the report “Future Food Systems: For people, our planet and prosperity” [23] GHG emissions could be reduced by 41–74% through a shift towards more healthy, sustainable diets. Such a change would also boost public health and increase resilience to climate shocks, which in turn would reduce the number of climate induced diseases and deaths.

## 6. Food Can Create Health or Unhealthiness—It Is a Matter of Culture

Unhealthy diets count for about one quarter of global deaths [49]. In September 2020 the Global Panel on Agriculture and Food Systems for nutrition released a foresight report “Future Food Systems: For people, our planet and prosperity” [23]. The report reveals that there are 690 million chronically undernourished people around the world and 11 million deaths per year can be linked to poor-quality diets. The progress to reduce malnutrition to achieve the SDG target is, according to the report, still too slow. In almost all countries, excess weight and obesity are increasing at the same time as undernutrition remains in the poor regions of the world. 

This inequality implies that a “one size fits all” solution is impossible, “there is no single dietary pattern that delivers ‘good health’ in every society” [23] (p. 32). According to the EAT-Lancet Commission summary report [12] for example, the consumption of red meat and livestock needs to be considered in its context. In some countries red meat is to be considered vital for a population suffering from undernutrition, while in wealthier countries there seems to be excessive consumption that could be reduced for health benefits both for the consumer and the planet. The consumption of ultra-processed foods has also increased worldwide [45]. Ultra-processed food is related to the occurrence of obesity and other non-communicable diseases and according to research, ultra-processed food also has a very high climate footprint (ibid.).

The overconsumption of food is considered to be a systemic failure of our modern food culture. In wealthy countries there is a food culture convincing the consumer that more is always better, but this stresses the intensification in agriculture and a trade where you constantly want to increase the volumes of food that are sold and consumed and which, unfortunately, is also often thrown away [6,50,51]. Low-income consumers often can’t afford, or don’t have access to healthy diets, while a great number of consumers who could afford a healthy diet still keep eating unhealthy food. This is due to a complex interaction between different factors such as cultural traditions, social norms, prices, marketing and environmental constraints [8]. The report “Future Food Systems: For people, our planet and prosperity” [23], also points out that the social and cultural norms also have a great impact on people’s beliefs around healthy diets. The report highlights that cultural norms around where people buy their food and where food is eaten, are changing. Family meals in the home are shifting towards a more frequent use of street foods, and eating at fast and full-service restaurants, and this has an impact on diet quality. Studies reveal that about 50% of sold meals in America are of poor or intermediate quality. The report [23] points out that fast-food restaurants don’t need to be worse than full-service restaurants in terms of diet quality, as a study of caloric content in full-service meals conducted randomly in different countries, revealed that the full-service meals on average contained about 33% more energy than fast-food meals. The public health community has long been aware of the negative impact from fast-foods and snacks on public health, but the report [23] emphasizes the important roles that the high energy content in restaurant meals play in the accelerating obesity epidemic. Sproesser et al. [51], argue that obesity has increased through increasing wealth, and might even affect more people in a couple of decades than food insecurity. According to Benton et al. [6] overeating is also a large contributor to food system losses.

“*…in high consuming countries, animal product consumption could be reduced and fruit and vegetable consumption increased, benefiting health in terms of reduced noncommunicable disease burden and reducing the impact on the environment…*”[52] (p. 2000)

In the poorest households’ wild foods are a source of food security. Wild food is often rich in micronutrients and has been shown to be vital for population groups living in poverty. For example, a study of rural Madagascar revealed that if access to wild meat was removed it would induce a 29 percent increase in the number of children suffering from anemia and a tripling of anemia cases among children. Wild foods contribute to food security, especially in connection with natural disasters and in times of scarcity. But wild food is also being collected and sold to provide an income, which might cause problems if it is not done in a sustainable manner. The issue is that overuse of wild products in some places causes implications for both biodiversity and the sustainability of the livelihoods of people relying on these resources. The use of wild foods can also, besides threatening biodiversity, cause other issues if it is incorrectly used, such as food safety and human health issues. Some OECD countries note that even though wild-food consumption is generally low, it still constitutes a substantial contribution to the diets for some population groups. For instance, in Finland the indigenous Sámi population continues to depend on wild fish and wild meat for a significant portion of its diet [14]. 

All over the world, we have different population groups with different cultural backgrounds and income levels, affecting their access to food and how they use available sources of food. In countries where undernutrition is common, red meat and wild food can save a population from malnutrition, but there are also countries with population groups that have an excessive consumption of red meat and ultra-processed food. Benton et al. [6] write about a global ‘double burden’ of malnutrition. This is due to the fact that the costs of producing and consuming meat have fallen, and because the food system is based on staple crops, that is cheaper calories, which have become increasingly abundant while nutritious crops have become more expensive and less available. This trend has led to an increase in overconsumption of calories, and underconsumption of nutrients, and in the long run an increase in non-communicable diseases [6,23]. The urbanization and globalization of the world, has in recent decades accelerated this process and the amount of obese and overweight people is rising in all countries around the world, both developed countries and developing, causing serious public health issues [53]. According to the WHO [7] non-communicable diseases kill 41 million people each year, equivalent to 74% of all deaths globally. 

Neufingerl and Eilander [43], found that a shift towards a plant-based diet and less use of animal foods could increase the intake of some important vitamins and other substances that may have benefits for health outcome, but the researchers argue that this kind of dietary shift needs to be carefully planned as there might be a risk of inadequate intake of other vitamins, trace elements and minerals vital to the human body and its function. They believe that consumers will need more help from public health strategies for a transition towards a sustainable and more nutritionally balanced diet. (Ibid.)

### A Scandinavian Example of a Traditional Food Culture

The Sami are Europe’s only indigenous people, and have their roots in northernmost Scandinavia and the Kola Peninsula in Russia [54]. The Sami are often described as the world’s healthiest indigenous people, which is unique from a global indigenous perspective. Research show that Sami men have a lower risk of dying from cancer than other men. Reindeer herding Sami men also have a lower risk of dying from cardiovascular disease and gastrointestinal diseases. A traditional Sami diet is primarily based on the animals and plants that belong in the northernmost Nordic region. In Sweden, this means a high daily intake of lake fish, blood, entrails and meat from reindeer, moose and other forest game, as well as various traditional plants such as berries, sorrel and fennel. Purchased goods such as coffee and flour have also had an important position in the traditional Sami household. Overall, this means a diet rich in fat and protein and poor in carbohydrates, fiber and certain vitamins. In other words, a traditional Sami diet differs markedly from established dietary advice, for example the Nordic nutritional recommendations [55]. 

## 7. Top Environmental Problems—A Result of Basic Values, Selfishness and Apathy

In September 2022 the UNPD published a report on SDG 2 end hunger: “Cultivating inner capacities for regenerative food systems. Rationale for action report” [8]. In the beginning of the first section of the report there is a quote from Gustave Speth, former Chair of the United Nations Development Group:

“*I used to think that the top environmental problems were biodiversity loss, ecosystem collapse and climate change. I thought with 30 years of good science we could address those problems. But I was wrong. The top environmental problems are selfishness, greed and apathy… And to deal with these we need a spiritual and cultural transformation*”.[8] (p. 10)

The report summarizes that although there have now been significant investments and efforts towards transforming the food systems, it appears the solutions emphasized so far are not efficient enough to deliver the necessary impact. This reveals the importance of looking at the root causes of our crises and finding the structural barriers to transformation, and these root causes are partly to be found within us humans. Until now the dominant idea in the community has been that we will find external solutions to the problems, disconnected from ourselves. Our cultural narrative of separation between humans and nature constitutes a fundamental reason to this approach [8].

This separation has evolved over centuries, ever since the scientific revolution and the spread of Christianity, which developed a philosophy that is dominating the western way of thinking today. This brings us back to the discussion on how different cultures embrace the concept of sustainability. In a previous section in this article, it is mentioned that the economic values of the European culture dominate the Western concept of sustainability [17]. We have created an economic and socio-political system based on the idea of an ever-growing production and consumption [56] and we see nature as an object that we can control through technological solutions [57,58,59]. Herman [57] argues that the disconnection between science, culture and traditional knowledge that the development has resulted in, is underpinning the problems we face today. According to research [8], it seems like inner capacities and systemic dysfunctions constitute an obstacle to achieving necessary change, and quite recently the discussion within the international development and sustainability area has increased its focus on human’s inner being as a part of the extensive problem humanity is facing (ibid.). Sustainability is an intricate concept, and it is strongly connected to fundamental values and ecological literacy [44].

In the last 25 years urbanization has led to a decrease in people’s understanding of ecological key concepts. The understanding of these ecological key concepts is fundamental for the development of the understanding of the intricate relationship between human systems and systems in nature. One of the most important ecological key concepts is sustainability, but unfortunately people’s ability to see and understand interconnections in natural systems is a disappearing knowledge. For example, the understanding of the importance of biodiversity for sustainability has often been shown to be incomplete and include misunderstandings [60,61,62,63,64]. Urbanization has increased the disconnection between humans and nature in a way that people often don’t see the relationship between food, the plants and animals and ecosystems that provide them with this valuable food [8]. Nowadays, most consumers are also disconnected from the food producer, as small local farms have had to give way to a more largescale intensified agriculture, we don´t know our local farmer anymore and we tend to value products from unknown producers to be less price worth [8]. Even the agriculture and farming production mechanisms are disconnected from nature as we keep using methods that harm the biodiversity, the soil, the water and the air [8]. This relates to Herman’s [57] thoughts about a disconnection from traditional knowledge. Hans Rosling [65], also argued that there is a gap between public and academic discourse. He argued that there is a lot of good science not coming to its right use as the gap between public and academic discourse is not bridged. He insisted that it is not the fault of the general population that there is a lack of popular knowledge, it is the researchers’ fault. To bridge this gap, Rosling argued that storytelling was an eminent tool. 

Bai et al. [66] also emphasizes the importance of an increased humanity and an increased understanding of the situation of others. Population groups with the most capacity, who can afford to consume and then also have shown to pollute the most, need to put in more effort to reverse climate change. Everyone needs to understand that it is the groups still living in extreme poverty, who strive to survive, that need to develop further and get access to products to consume for their survival. This while groups with excessive consumption could advantageously reduce their consumption. 

There seems to be some consensus around the idea that a change is needed in our basic values to re-evaluate the current philosophy where humans are disconnected from nature. But values do not change in the blink of an eye. One of the tools that could be a cornerstone in reconnecting humans to nature and culture is education, but that presupposes first and foremost a changed education because it is partly the education we have today that has taken us to this state of urgently needed changes [67]. 

The adaptation towards a transformative learning approach in education could contribute to educating engaged and sustainability conscious society members [68] but the implementation of changes in education is known to be time-consuming. That is why we also need other areas with change agents [67]; e.g., informal education could be a good and effective way to spread knowledge, transformative tourism could be another way to educate and inform people. We will need to find possible areas and tools in every field we can think of to bring about change fast enough to secure our food systems, stop the loss of biodiversity and to slow down climate change. Both innovations and a change in our mindset will be crucial to manage the needed change. Once again, we will urgently need and are dependent on talented consumerism architects that know how to shape norms, values and narratives. 

### The Headlines Project in Finland—A Part of the World’s Largest Food Innovation Community

At Centria University of Applied Sciences in Kokkola, Finland, there are a couple of food research projects. For example, the HEADLINES project, which is mainly coordinated by EIT Food, a pan-European organization supported by the EU. The goal of EIT food is to accelerate innovation to build a future-fit food system that produces healthy and sustainable food for all, with the main educational goal to attract, develop and empower bright minds to lead the transformation of the food system to become an innovative sector, trusted by society, that produce healthy and sustainable food [69,70]. 

The HEADLINES project [71] is working diligently to place a stronger emphasis on supporting companies and emerging entrepreneurs in the food and healthcare sectors. The project focuses on incrementally improving the degree to which entrepreneurship is included within the core ecosystems of the partner universities. All partners in the project are deeply involved in promoting and developing entrepreneurial activity. The project offers a platform for collaboration and enhances the opportunities to work together in sharing research, background experiences and good practice knowledge. It also promotes the development of new shared knowledge and well-focused policies on deeply embedding entrepreneurship into our educational systems. Although entrepreneurship is the main goal of the project, it still has its activities turned towards entrepreneurship and innovation for the food and healthcare sectors. 

The present article is partly written to support the activities of the HEADLINES project and the EIT food goals, as it focuses on both food systems and health research and provides insights concerning a transition towards a food and agricultural system that could stop the global loss of biodiversity. 

## 8. Summary and Future Research

This article reveals that the problem with our current food system is multifaceted. It is a tremendous system that has both a positive and a negative impact on people, plants, animals and climate. It contains and affects factors at different levels, such as greenhouse gas emissions, climate change, biodiversity loss, food systems, production, access malnutrition, overeating, diseases and health and all these components among a variety of others are interconnected in an invisible web. It appears quite obvious that there is a need to include multiple disciplines to find solutions and pathways forward to achieve a sustainable and secure food system.

Many policy documents and researchers advocate a dietary change to reduce the pressure on nature, reduce greenhouse gases and at the same time improve the health of the population. To start a process towards a dietary change all available facts from different disciplines must be discussed and put on the table. For example, the pros and cons of different solutions and agricultural techniques need to be carefully considered. It is highly important that different disciplines start working together to find the best solutions and pathways forward. Innovations are seldom born among like-minded, dissenters and multidisciplinary groups are required. These kinds of pervasive changes are not easy to implement, but examples show that it is possible to transform the demand and supply with the right means.

There is a need to re-evaluate and review our current approach and mindset concerning food and our food culture. If we decide not to eat meat or reduce meat eating, we might need to consider if we really need ultra-processed substitutes instead. If we don’t eat sausage we might need to consider as a consumer if we really need veggie sausage. Future research is needed to clarify the impact of ultra-processed products and the agricultural system on human health and biodiversity. At the same time research is needed to understand and develop knowledge around sustainable norm shaping, value shaping and how to create real narratives to raise awareness about the impact of consumption on our environment.

This article highlights three identified gaps that need to be bridged for a sustainable future. The gap between humans and nature, the gap between science and culture, and the gap between academic and public discourse. The first two mentioned are deeply rooted within humans. Transformation, education and innovation in different disciplines will be a necessity for a secure and sustainable food system in the future, but in the end it all comes down to fellow humanity, empathy and morals, give and take, sharing. Everyone needs to understand that health is wealth.

## Data Availability

Data is contained within the article.

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
