# Peer review of "A Transition towards a Food and Agricultural System That Includes Both Food Security and Planetary Health"

_foods, 2022, doi:10.3390/foods12010012_

Round 1
Reviewer 1 Report
The paper raises a very important problem on the global level in a descriptive way. It provides a general, qualitative literature review, which is a value it itself. It seems to me it should be classified as Review rather than Article, as no own empirical research data are reported.
I would appreciate also a section analysing the strenghs and weaknesses of the extant literature of the subject, with clear future research directions.
Incomplete reference: 54
The quality of English is rather low. The paper should undergo English proofreading. Selected suggestions for improvement below:
line 17 - is, which
18 - identifies
43 - 2.4
47 - obesity
87 - seems
95 - material
111 - increases
119 - reduce
208 - remove "does"
260 - has
267 - suggest
321 - increased
337 - reveal
344 - emphasizes
362 - remove "as"
371 - write
412 - "although" instead of "despite"
421 - Christianity,
430 - constitute
431 - discussion
446 - have
459 - have
491 - focuses
494 - enhances
517 - need
Author Response
According to the statement that it should be classified as Review rather than Article, as no own empirical research data are reported:
A theoretical article don´t need own empirical research. There are lots of theoretical articles in different scientific papers with no own empirical research.
I changed all the misspellings reviewer one pointed at line by line.
For the incomplete reference: 54, I wrote the English title within [ ] as it is an article in Swedish
Reviewer 2 Report
The author presents a critical review about food systems' sustainability, putting into perspective some figures and pointing out misconceptions. The main message is the need for multidisciplinary research to tackle complex interrelated issues. The work is written in simple yet rigorous language. Minor changes in adjusting structure and spelling are suggested below:
line 33 - "presentation presented" - pls rephrase
l.47: please replace 2014 by 2016, according to the update of your reference: https://www.who.int/news-room/fact-sheets/detail/obesity-and-overweight
l. 67-69 should be moved up because the aim of the critical review is expected to be placed by the end of the introduction section (maybe before the paper's "roadmap")
l. 70-75 can be deleted and the method' section may start at " This study is based on..."
l. 82. please provide one or more references to allow the reader exploring the methodology used in the current critical review
l.83-88 are considerations and therefore do not fit in this section which should only describe research procedures/methodologies in a pragmatic objective way. Pls rephrase or move the paragraph elsewhere.
l. 95 misspelling - pls rewrite
l. 96. Primary sector activities such as agriculture, forestry and fisheries...
l.106 and others - pls replace man/mankind by human/humankind along the manuscript
l. 109 pls delete "is"
l. 111-112: this is totally true for higher species but microbes, including human and crop pathogens (bacteria, fungi, viruses), adapt really fast because of their short generation times (of hours); there are gaps in knowledge in this respect, but you may wish to discuss it because it is again mentioned below - you may wish to note the evolutionary advantages of some undesired species favoured by unbalanced - as invasive animals and plants and human and crop pathogens (which are parasites)
l. 197 as the Finnish
l.227 long turn or long run?
l. 287 Verdi et al. [47]
l. 359. you may wish to consider that the vulgarization of wild foods, besides the threat to biodiversity, may also pose serious food safety & human health issues - covid19 originated in a wet food market with wild animal foods
l. 423 in a previous section (....) it is mentioned
Congrats on the good work
Author Response
According to reviewer 2:s suggestions I rephrased the sentence in (originally line 33) - "presentation presented"
l.47: please replace 2014 by 2016, according to the update of your reference.
- Done
67-69 should be moved up because the aim of the critical review is expected to be placed by the end of the introduction section (maybe before the paper's "roadmap").
- Done
70-75 can be deleted and the method' section may start at " This study is based on...".
- Done
82. please provide one or more references to allow the reader exploring the methodology used in the current critical review.
- Done
l.83-88 are considerations and therefore do not fit in this section which should only describe research procedures/methodologies in a pragmatic objective way. Pls rephrase or move the paragraph elsewhere.
- Moved to other section.
95 misspelling - pls rewrite
- Done
96. Primary sector activities such as agriculture, forestry and fisheries...
- Done
l.106 and others - pls replace man/mankind by human/humankind along the manuscript
- Done
109 pls delete "is"
- Done
111-112: this is totally true for higher species but microbes, including human and crop pathogens (bacteria, fungi, viruses), adapt really fast because of their short generation times (of hours); there are gaps in knowledge in this respect, but you may wish to discuss it because it is again mentioned below - you may wish to note the evolutionary advantages of some undesired species favoured by unbalanced - as invasive animals and plants and human and crop pathogens (which are parasites)
- Done briefly
197 as the Finnish
- Done
l.227 long turn or long run?
- Changed to long run
287 Verdi et al. [47]
- Done
359. you may wish to consider that the vulgarization of wild foods, besides the threat to biodiversity, may also pose serious food safety & human health issues - covid19 originated in a wet food market with wild animal foods
- Done briefly
423 in a previous section (....) it is mentioned
- Done